# Influence of the Elastic Modulus on the Osseointegration of Dental Implants

**DOI:** 10.3390/ma12060980

**Published:** 2019-03-25

**Authors:** Aritza Brizuela, Mariano Herrero-Climent, Elisa Rios-Carrasco, Jose Vicente Rios-Santos, Roman A. Pérez, Jose Maria Manero, Javier Gil Mur

**Affiliations:** 1Department of Prosthodontics and Occlusion, School of Dentistry, University of Oviedo, 37006 Oviedo, Spain; aritzabrizuela@hotmail.com; 2Faculty of Odontology, University of Seville, 41009 Seville, Spain; mariano@herrerocliment.com (M.H.-C.); elisarioscarrasco@gmail.com (E.R.-C.); jvrios@us.es (J.V.R.-S.); 3Institute of Bioengineering, School of Dentistry, Universitat Internacional de Catalunya, 08195 Barcelona, Spain; rperezan@uic.es; 4Biomaterials, Biomechanics and Tissue Engineering Group, Department of Materials Science and Metallurgy, Technical University of Catalonia (UPC), EEBE, 08019 Barcelona, Spain; jose.maria.manero@upc.edu

**Keywords:** low elastic modulus, dental implant, osseointegration, titanium alloys

## Abstract

The load transfer from metallic prosthesis to tissue plays an important role in the success of a designed device. From a mechanical behavior point of view, the load transfer will be favored when the elastic modulus between the metallic implant and the bone tissue are similar. Titanium and Ti-6Al-4V are the most commonly used metals and alloys in the field of dental implants, although they present high elastic moduli and hence trigger bone resorption. We propose the use of low-modulus β-type titanium alloys that can improve the growth of new bone surrounding the implant. We designed dental implants with identical morphology and micro-roughness composed of: Ti-15Zr, Ti-19.1Nb-8.8Zr, Ti-41.2Nb-6.1Zr, and Ti-25Hf-25Ta. The commercially pure Ti cp and Ti-6Al-4V were used as control samples. The alloys were initially mechanically characterized with a tensile test using a universal testing machine. The results showed the lowest elastic modulus for the Ti-25Hf-25Ta alloy. We implanted a total of six implants in the mandible (3) and maxilla (3) for each titanium alloy in six minipigs and evaluated their bone index contact (i.e., the percentage of new bone in contact with the metal—BIC%) after 3 and 6 weeks of implantation. The results showed higher BIC% for the dental implants with lowest elastic modulus, showing the importance of decreasing the elastic modulus of alloys for the successful osseointegration of dental implants.

## 1. Introduction

At present, a major concern of metallic prosthesis implanted in the body (e.g., dental implants) is the stress-shielding effect, which is due to the significant differences between the Young’s or elastic modulus of bone tissue and those of the metallic prosthesis. When a load is applied, the material with the highest elastic modulus—generally the metallic prosthesis—will adsorb the load, hence hindering bone regeneration while enhancing bone resorption [1,2]. In order to overcome this issue, new β titanium alloys that present similar mechanical properties to those of bone are currently being developed [3]. The most common mechanism consists of using biocompatible β-stabilizing alloying elements such as tantalum, niobium, molybdenum, hafnium, and sometimes also zirconium, because the β-phase titanium alloys show lower elastic modulus compared to the α-phase ones [4,5]. Several research projects have concentrated on the production of natively biocompatible alloys, such as the Ti–Nb and the related Ti–Nb–X system (where X = Zr, Ta, Au, O). These alloys have yielded high superelastic strains (i.e., as high as 4.2%), while avoiding biocompatibility issues [6,7,8]. In a similar way, we developed the TiNbHf ternary alloy presenting very interesting properties with low elastic modulus [9,10,11,12], ideal for fabricating orthopedic implants. While we have previously described the fabrication of several alloys with lower elastic modulus, an in vivo experimental design still needs to be performed to prove their use as load-bearing implants. The objective of the present work was to validate the properties of the novel low elastic modulus alloys in terms of their mechanical behavior and their in vivo osseointegration.

## 2. Results and Discussion

A major concern of implants that substitute hard tissues is the poor match in terms of mechanical properties between the bone and its substitute. This effect is known as stress shielding, and has been associated with severe bone mass loss. Cortical bone has an elastic modulus of 20 GPa, whereas the most common commercial alloys have an elastic modulus ranging from 110 GPa for Ti6Al4V to 220 GPa for the cobalt-chrome alloys. In order to reduce the stress-shielding effect, several efforts have been directed towards finding strategies to reduce the elastic modulus. For instance, we previously showed a reduction of the elastic modulus by increasing the plastic deformation by means of thermo-mechanical treatment [13,14]. Although the value achieved was higher than the Young’s modulus of compact (cortical with high density) bone, this alloy is potentially a very interesting candidate for load-bearing implant applications.

The alloys described in the current work are based on Ti doped with different amounts of Zr, Nb, Hf, or Ta, since these are considered excellent β-phase stabilizers and hence are able to reduce the elastic modulus and increase the ductility of the alloys. The common principle of these alloys is that they promote good workability at cold temperatures, which allows their mechanical properties to be modified through microstructural changes [15,16], reaching values for elastic modulus as low as 40 GPa [3,5,17]. The results of our tensile tests are shown in Table 1. The pure Ti (grade 2) and the Ti6Al4V alloy presented values of 107 and 113 GPa respectively, not showing significant differences among them. The incorporation of Zr (Ti-15Zr) represented a very slight reduction of the elastic modulus compared to the Ti cp, having a value of 103 GPa not presenting significant differences. Nevertheless, Ti was then combined with Zr and different Nb doping, showing significant reductions in the elastic modulus, presenting 74 and 67 GPa values for Ti-19.1Nb-8.8Zr and Ti-41.2Nb-6.1Zr, respectively. Finally, the incorporation of both Hf and Ta also showed a significant reduction in the elastic modulus value, reducing it to 53 GPa.

The different alloys were machined into dental implants to perform an in vivo study. Figure 1 shows the dental implant machined with the described alloy (Klockner SA, Escaldes Engordany, Andorra). A shot blasting treatment was performed in order to provide the implants with surface roughness and hence to promote the bonding of the bone with the implant.

The surface roughness created by the shot blasting was analyzed by interferometric analysis. Table 2 summarizes the surface roughness values for each sample. The different Ti-based alloys presented similar roughness values to pure titanium, not presenting significant differences among them. The shot blasting process generated surface roughness with values of Ra ≈ 2 µm, which were similar to those reported in the literature [18]. Similarly, the Pc values did not present significant differences, although the values showed slightly more distant values, presenting values around 80 cm^−1^. The absence of significant differences among the surface roughnesses of the different alloys allowed the surface roughness to be removed as a variable that could have an effect on the osseointegration process, and was therefore considered to be constant in all the different tested alloys.

Finally, an in vivo implantation in the maxillae of minipigs was performed. At 3 and 6 weeks after implantation, animals were sacrificed and histological sections were stained in order to analyze bone growth and bone-to-implant contact. The different staining for the interface between the bone and the implant for the different alloys are shown in Figure 2.

The new bone formed is shown in the bright purple color, whereas old bone is shown in light purple. As can be observed, the Ti cp and Ti6Al4V presented low densities of new bone in contact with the dental implant. The amount of new bone in contact with the implant was shown to be similar in the Ti-15Zr. Nevertheless, these areas of new bone increased in the presence of Nb at 19% and 41.2% content. Finally, the Ti-25Hf-25Ta showed significant amounts of new bone surrounding the bone implant, which was then correlated with the higher BIC values.

The results of the BIC at 3 and 6 weeks are shown in Figure 3. The general trend showed that there was an increase in the BIC values when comparing 3 and 6 weeks as expected. However, the Ti-6Al-4V implants after 6 weeks presented BIC values without statistically significant differences in relation to the Ti cp (*p* > 0.15) and Ti-15Zr (*p* > 0.19). Ti cp, Ti-15Zr, and Ti-19.1 Nb-8.8Zr alloys did not present statistically significant differences after 6 weeks of implantation. The alloys with lowest elastic modulus (i.e., Ti-41.2Nb-6.1Zr and Ti-25Hf-25Ta) presented statistically significant differences in relation to the Ti cp, Ti-15Zr, and Ti-19.1 Nb-8.8Zr with *p*-values < 0.005. Interestingly, the BIC% increased as the elastic modulus decreased. This was related to the reduction of the elastic modulus, which allowed a greater and more efficient transfer of the mechanical load to the bone, favoring the formation of new bone around the dental implant. It has been previously reported that lower elastic modulus has also shown better dental implant osseointegration capacities [19]. Furthermore, the combination of adequate topographical cues with low elastic modulus has been shown to present optimum osseointegration results [20]. Moreover, the biocompatibility of these systems has been evaluated, showing excellent results, with no evidence of ion release after 4 weeks in vivo [21] and excellent corrosion behavior. Therefore, properties such as high maximum and proof strength, low elastic modulus, potential supereleasticity, and excellent biocompatibility make this alloy a real candidate for many applications.

These alloys have been studied in terms of their corrosion, ion release, and biocompatibility in vitro with human osteoblastic cells [22,23], showing excellent behavior as biomaterials. However, it is also necessary to evaluate their antigenicity, toxicity, and allergenicity.

## 3. Materials and Methods

The designed alloys containing Ti, Nb, Hf, Ta, and Zr were obtained by mixing the corresponding amounts of each component in an arc melting furnace. The initial reagents were obtained from Fort Wayne Metals Company as follows: Ti cp bars (grade 2), Nb foil (99.8% purity), Hf shavings (99.7%), Ta powder (99.2% purity), and Zr bars (99.8% purity). Table 1 shows the chemical composition of the different alloys obtained by means of X-ray dispersive energy EDS Oxford 240 (Oxford Instruments, Abingdon-on-Thames, UK).

In order to quantify the mechanical properties of the designed alloys, tensile assays were performed for each group. In each case, ten tensile specimens with a diameter-to-gauge length ratio of 1/5 were machined. The tests were performed according to the ASTM-E8-04 (18) standard [18]. These were tested at a cross-head displacement rate of 1 mm/min in a universal testing machine (Bionix 358, MTS, Eden Prairie, MN, USA).

In order to prove that the different alloys could be topographically modified in order to confer better osseointegration properties, the implants were shot blasted with Al_2_O_3_ to modify the surface roughness of the implants. Roughness was evaluated by white light interferometer microscopy (NT1100 Wyko, Veeco, Plainview, NY, USA). The surface analyzed was 189 × 248 mm^2^ for the smooth control surfaces and 460 × 604 mm^2^ for all the rough surfaces. Data analysis was performed with 232TM Wyko Vision software (Veeco, Plainview, NY, USA). A Gaussian filter was used to separate the waviness and morphology from the roughness. The roughness values were measured in four different places of each type of dental implant to characterize the amplitude and spacing roughness parameters Ra and Pc. Ra is the arithmetic average of the absolute values of the distance of all points of the profile to the mean line. Pc is the number of peaks in the profile per length of analysis.

In order to verify the different abilities of the implants to osseointegrate, an in vivo animal experiment was performed. The study was conducted with a total of six 6-year-old female minipigs in accordance with the Declaration of Helsinki, and the protocol was approved by the Ethics Committee of University of Cordoba (Spain) Project number 67183-18. The dental implants were inserted into the maxilla and mandible following statistical control. Five dental implants of each alloy were inserted into the maxilla and five into the mandible. In all cases, the implants were loaded with their abutments in order to simulate the real behavior. The dental implants presented identical design and measurements and had external connection with hexagonal shape.

Four months before the surgical implantation of the dental implants, all the mono- and multiradicular teeth were extracted. Minipigs were initially preanesthetized with ketamine and xylazine and maintained on gas anesthesia (5% isoflurane/O_2_). The minipigs were continuously infused with lactated Ringer solution to keep animals hydrated. The effect of anesthesia was continuously monitored by tracking changes in the heart rate, the depth of respiration, as well as the respiratory rate. In order to further verify their level of awareness, the minipigs were frequently toe-pinched and tested for corneal reflex.

Regarding the implantation procedure, a semi-submerged technique was performed. Implants were inserted in the maxilla and mandible of the minipigs. After implanting the dental implants in the bone bed, flap margins were adapted and tension-free sutured using bioresorbable polyglictin (910-Vycril^©^ 3-0, Ethicon, Somerville, MA, USA). After surgery, amoxicillin was administered for infection control and buprenorphine HCl for pain control. The animals were monitored for swelling, dehiscence, and infection.

Each animal received five implants with each of the alloys and were sacrificed after 3 and 6 weeks after surgery. Different in vivo studies in minipigs have shown that 6 weeks is sufficient to complete the osseointegration process [18,21].

After euthanasia, thick block sections of 16 mm including implants, alveolar bone, and surrounding mucosa were collected by cutting with an irrigated diamond-saw (Accutom 50, Struers, Tokyo, Japan) and radiographed. The samples were thoroughly rinsed in sterile saline solution and immersed in buffered 10% formol solution. The tissue blocks were fixed, dehydrated in ethanol solutions, and embedded into molds in a photopolymerizable resin according to the method of Gil et al. [24].

The embedded implants were cut midaxially in a buccal–lingual plane into sections approximately 200-mm thick. They were further treated to obtain a final polished section that was 50 µm thick. Sections were then stained with toluidine blue for 20 min following the methodology of Reference [22]. The histopathologic and histometric analyses were performed with a digital camera system (DP12, Olympus, Tokyo, Japan) attached to a light microscope (BX51, Olympus) and image analysis software (MicroImage 4.0, Olympus). Osseointegration of the dental implants was assessed by calculating the percent of bone–implant contact (BIC) along the total length covered by the pictures.

Statistically significant differences among test groups and control for both elastic modulus and histometry were assessed using statistical software (Minitab^TM^ 13.1, Minitab Inc., Fort Wayne, IN, USA). ANOVA tables with multiple comparison Fisher test were calculated. The level of significance was established at *p*-value < 0.05.

## 4. Conclusions

The elastic modulus of titanium and titanium base alloys have a great influence on osteointegration. All the alloys studied presented a good osseointegration for the times tested. We proved greater values of osseointegration were observed with dental implants having abutments fabricated with an elastic modulus close to that of bone. Hence, titanium beta alloys are excellent candidates for this improved biological response with higher osseointegration rates.

## Figures and Tables

**Figure 1 materials-12-00980-f001:**
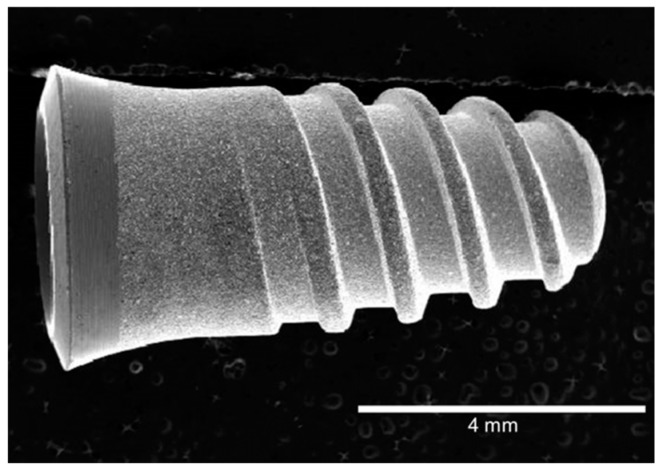
Morphology of the dental implant (Klockner SA).

**Figure 2 materials-12-00980-f002:**
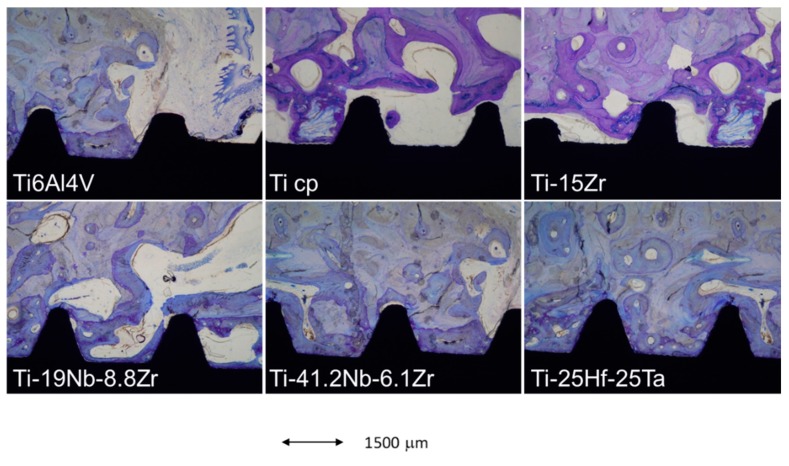
Toluidine blue histological staining of the tissue sections retrieved after 6 weeks for the different implanted alloys, showing old (light purple) and new bone (bright purple), and the dental implant (black).

**Figure 3 materials-12-00980-f003:**
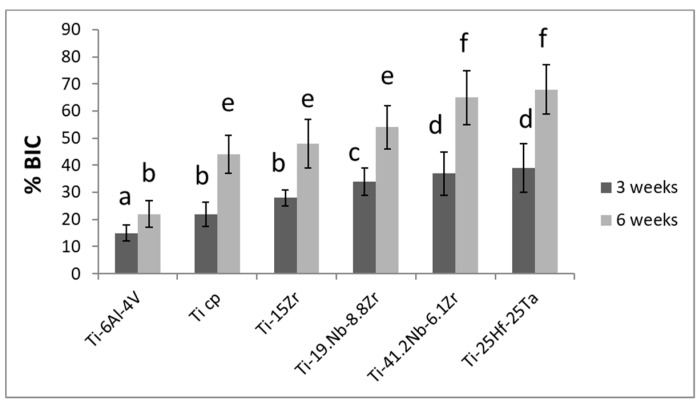
Bone-to-implant contact (BIC) for the different alloys after implant retrieval at 3 and 6 weeks after implantation. Columns labeled with the same letter are not significantly different.

**Table 1 materials-12-00980-t001:** Mechanical properties in tensile test.

Ti alloys	E (GPa)
Ti-6Al-4V	113 ± 3
Ti cp	107 ± 3
Ti-15Zr	103 ± 2
Ti-19.1Nb-8.8Zr	74 ± 2
Ti-41.2Nb-6.1Zr	67 ± 2
Ti-25Hf-25Ta	53 ± 3

**Table 2 materials-12-00980-t002:** Roughness of the different dental implants.

Ti Alloys	Before	Shot Blasting	After	Shot Blasting
	Ra (µm)	Pc (cm^−1^)	Ra (µm)	Pc (cm^−1^)
Ti-6Al-4V	0.56 ± 0.14	8.2 ± 0.3	1.99 ± 0.41	98.3 ± 2.4
Ti cp	0.43 ± 0.11	9.9 ± 1.2	2.33 ± 0.54	70.9 ± 9.2
Ti-15Zr	0.33 ± 0.07	8.4 ± 1.7	2.23 ± 0.76	88.1 ± 11.1
Ti-19.1Nb-8.8Zr	0.44 ± 0.10	7.3 ± 1.6	1.74 ± 0.25	82.1 ± 10.0
Ti-41.2Nb-6.1Zr	0.33 ± 0.07	8.8 ± 1.2	1.83 ± 0.33	92.1 ± 13.0
Ti-25Hf-25Ta	0.23 ± 0.05	6.9 ± 2.2	2.33 ± 0.53	70.9 ± 9.2

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
