# Peer review of "Influence of the Elastic Modulus on the Osseointegration of Dental Implants"

_materials, 2019, doi:10.3390/ma12060980_

Round 1
Reviewer 1 Report
This manuscript, entitled 'Influence of the elastic modulus on the osseointegration of the dental implants,' is considered to be within the scope of this journal. Before it is acceptable for publication, there are some critical issues to be addressed.
This study has not contained any in vitro results of the tested new biomaterials, which will be inserted into the human bone. The tests are necessary for biocompatibility, antigenicity, toxicity, and the like. At least, this limitation of this study should be described in the 'Results and Discussion' section.
The authors describe that the BIC% increased as the elastic modulus decreased. Which results have made the authors think so? The specimen implants were inserted into the maxilla or mandible of the experimental animals, minipigs. Did the authors deliver implant-supported restorations connected to the inserted implants? If not, the reviewer thinks that there would be no condition where the effect of the elastic modulus was shown. There was another critical variable that had an effect on BIC; surface chemistry. Did the authors investigate the surface chemistry of the groups using XPS or EDS? Also, 3 and 6 weeks for sacrifice were considered to be too short to evaluate the effect of the elastic modulus of the test materials. How fast is the metabolic rate of the minipig?
Several mistakes have to be corrected.
Figure 2: scale bars must be inserted.
Ti cp: which grade of commercially titanium was used in this study?
Short description of the surface parameters, Ra and Pc, is needed.
Figure 3: significant differences should be marked.
Significant differences should be described in 'Results and Discussion' with p-values.
Author Response
Dear Reviewer,
Thanks for taking the time to review our manuscript and suggest to us to improve our work by providing a lot more detail. We have done so, and we are now submitting a manuscript that not only addresses the points the you specifically raised but also many others that we have considered in order to deliver what we think is a much improved version of our work. This version includes more paragraphs in all main sections, new tables, new results to better reflect the contents of our contribution. There are large number of changes and so, we have not specifically highlighted all of them.
We are looking forward to your comments.
Sincerely,
Francisco-Javier Gil Mur
Reviewers' comments:
Reviewer #1:
1.The tests are necessary for biocompatibility, antigenicity, toxicity. At least, this limitation of this study should be described in the Results and Discussion section.
The authors have realised this description of the limitations and they have incorporated the references about these tests.
2. The results taht the BIC increased as the elastic modulus decrease. These results are demonstrated with the histological studies and the mechanical properties. The statistical studies have demonstrated the statistical significant differences.
3. Experimental method about in vivo test has been explained with more details.
4. Surface characterisation has been realised by means EDS.
5. Different references have been incorporated about the 3 and 6 weeks for sacrifice the minipigs. In these references are showed the test times for the histology.
6. In the Figure 2 scale bar has been inserted.
7. The grade of CP Ti has been incorporated in the text.
8. The parameters Ra and Pc have been described in the text.
9. Significant differences have been marked in the Figure 3 and in the text with p-values.
Reviewer 2 Report
please see attachment

Author Response
Dear Reviewer,
Thanks for taking the time to review our manuscript and suggest to us to improve our work by providing a lot more detail. We have done so, and we are now submitting a manuscript that not only addresses the points the you specifically raised but also many others that we have considered in order to deliver what we think is a much improved version of our work. This version includes more paragraphs in all main sections, new tables, new results to better reflect the contents of our contribution. There are large number of changes and so, we have not specifically highlighted all of them.
We are looking forward to your comments.
Sincerely,
Francisco-Javier Gil Mur
1. In the Figure 2 scale bar has been inserted.
2. The roughness and their parameters have been explained with more details in the text. The differences are very small and do not present statistical significant differences. The authors have been incorporated the roughness values before sand blasting treatment in the Table.
3. The relationship between elastic modulus and bone index contact have been revised. The statistical significant differences have been incoporated in the Figure 3 and the text with p-values.
4. In vivo test has been explained with more details. The implants were inserted in the different bone types in the minipigs –maxila and mandible- in the same number for each titanium alloy.
5. The dental implants have in all cases the abutment made of the same alloy in order to favour the load transfer.
Reviewer 3 Report
Comments:
There is no explanation of the abbreviations used – separately (BIC, LEM, etc.)
In the Introduction: A better explanation is required of the purpose of the contribution – with a more scientific approach
Change: first Materials and Methods as Part 2 and then Results and Discussion as Part 3
Improvement in presentation of Materials and Methods is needed (too confusing!)
Figure 2 – a magnification scale is needed, and an explanation is necessary regarding the display in the picture (what is it – the black and white part?)
The text has to be checked by a »native« speaker
Note the spaces when quoting references
Improvement is needed in the Conclusions Part
Author Response
Dear Reviewer,
Thanks for taking the time to review our manuscript and suggest to us to improve our work by providing a lot more detail. We have done so, and we are now submitting a manuscript that not only addresses the points the you specifically raised but also many others that we have considered in order to deliver what we think is a much improved version of our work. This version includes more paragraphs in all main sections, new tables, new results to better reflect the contents of our contribution. There are large number of changes and so, we have not specifically highlighted all of them.
We are looking forward to your comments.
Sincerely,
Francisco-Javier Gil Mur
1. The abbreviations have been explained.
2. The introduction has been improved with new text.
3. Materials and Methods is after of the Result and Discussion part following the authors guidelines of this Journal.
4. In the Figure 2 scale bar has been inserted. Black part is the titanium.
5. The text has been revised by a native speaker.
6. The conclusions have been improved.
7. Spaces when quoting references has been considered.
Round 2
Reviewer 1 Report
It is now considered to be acceptable for publication of this journal.
Reviewer 2 Report
OK! No further question!
Reviewer 3 Report
no